# Multimodal analgesia in resource-limited settings: A comparative analysis of postoperative pain management strategies in Pakistan

Sana Wazir[1], Summaya Inayat[2], Muhammad Jawad Ullah[3], Gulmakay Zaman[4], Abdur Rehman[5], Faisal[6], Shallozan[5], Nuzhat Rahil [7]*

1 Department of Oral and Maxillofacial Surgery, Medical Teaching Institute, Lady Reading Hospital, Peshawar, Pakistan, 2 Department of Chemistry, Shaheed Benazir University, Peshawar, Pakistan, 3 Department of Allied Health Sciences, Iqra National University, Peshawar, Pakistan, 4 North West School of Medicine, Peshawar, Pakistan, 5 Department of Gynecology, Hayatabad Medical Complex, Peshawar, Pakistan, 6 Kabul University of Medical Sciences, Kabul, Afghanistan, 7 Department of Ophthalmology Medical Teaching Institute, Lady Reading Hospital, Peshawar, Pakistan

* nuzhatrahilophthalmology@gmail.com

## Abstract

Postoperative pain remains a critical yet often under-managed aspect of surgical recovery, influencing patient satisfaction, functional recovery, and quality of care. This study aimed to evaluate the comparative analysis of different postoperative pain management strategies and their impact on patient-reported outcomes in a tertiary care hospital in Pakistan. A cross-sectional analytical study was conducted among 431 surgical patients at a tertiary care facility. Data were collected on demographic variables, type of surgery, pain management strategy (monotherapy, opioid-based, multimodal, adjuvant), route of administration, analgesic frequency, patient satisfaction, side effects, and functional interference. Statistical analyses including chi-square tests, ANOVA, Pearson's correlations, were performed. A p-value < 0.05 was considered statistically significant. Multimodal therapy was the most used strategy (29.9%), followed by opioid-based (27.6%) and monotherapy (26.7%). Patient satisfaction was significantly associated with pain management strategy (p < 0.001), with 36.9% reporting satisfaction and 20.2% very satisfied. While multimodal therapy showed lower mean pain scores than opioid-based regimens, the difference between these two groups was marginally non-significant (p = 0.060), suggesting a trend but not definitive superiority in terms of pain reduction. However, multimodal therapy was significantly associated with earlier mobilization compared to opioid-based regimens (p < 0.001). Significant correlations were observed between type of pain management and both pain score (r = -0.177, p = 0.000) and mobilization time (r = -0.116, p = 0.016). Preoperative anxiety and depression were present in 29.9% of participants and significantly associated with pain perception (p = 0.005). Multimodal pain management significantly improves postoperative outcomes by reducing pain intensity and

**Data availability statement:** The datasets generated and/or analyzed during the current study are included in this manuscript.

**Funding:** The author(s) received no specific funding for this work.

**Competing interests:** The authors have declared that no competing interests exist.

enhancing functional recovery. Implementing standardized multimodal analgesia protocols may optimize care quality in surgical patients.

## Introduction

Postoperative pain remains a significant clinical challenge, with studies estimating that 20–40% of patients worldwide experience inadequate pain relief following surgery, leading to prolonged recovery, increased healthcare costs, and higher risks of chronic pain development [1–3]. While various pain management strategies exist, opioid-based regimens continue to dominate clinical practice despite their well-documented adverse effects, including nausea, respiratory depression, and potential for dependence [4,5]. In recent years, multimodal analgesia - combining opioids with non-opioid adjuvants like NSAIDs, gabapentinoids, and regional anesthesia techniques - has emerged as the gold standard in high-income countries, demonstrating superior pain control with reduced opioid requirements [6,7]. However, critical gaps remain in our understanding of how these approaches translate to low- and middle-income countries (LMICs), where resource constraints, cultural differences in pain perception, and variations in healthcare infrastructure may significantly impact outcomes [8–10].

The current literature exhibits several important limitations. First, most evidence supporting multimodal analgesia comes from Western populations, with minimal data from South Asian contexts where genetic, cultural, and pharmacological factors may alter treatment responses [11–13]. Second, few studies have examined the implementation challenges of multimodal approaches in resource-limited settings, where medication availability, cost considerations, and training limitations may affect feasibility. Third, while psychological factors like preoperative anxiety and depression are known to influence pain perception, their impact has been understudied in LMIC surgical populations. Finally, existing research predominantly focuses on short-term pain scores rather than functional recovery metrics or long-term outcomes like chronic pain prevention.

This study addresses these gaps by comprehensively evaluating postoperative pain management strategies in a Pakistani tertiary care center. We compare the effectiveness of four approaches - monotherapy, opioid-based regimens, multimodal analgesia, and adjuvant therapies - across multiple outcomes: pain intensity, functional recovery, patient satisfaction, and side-effect profiles. Our work represents the first systematic assessment of multimodal analgesia in this context while incorporating novel elements like gender-specific analyses and psychological factor assessments. By providing LMIC-specific evidence, we aim to inform the development of context-appropriate pain management protocols that balance efficacy, safety, and feasibility in resource-constrained environments. We hypothesize that multimodal analgesia will demonstrate superior outcomes compared to traditional opioid-centric approaches, with variations influenced by surgical type, psychological factors, and patient demographics.

Our findings have important implications for clinical practice and health policy in LMICs. As global healthcare systems strive to reduce opioid dependence while

improving pain management, this study provides crucial data to guide protocol development in under-resourced settings. Furthermore, by identifying local barriers to effective pain control and highlighting population-specific response patterns, our work contributes to the growing movement toward personalized, context-sensitive postoperative care strategies. The integration of psychological assessments with clinical outcomes offers additional insights that could transform preoperative evaluation processes in similar healthcare environments worldwide. The primary research question guiding this study was: Among surgical patients in a resource-limited setting, how are postoperative pain management strategies associated with pain intensity and mobilization time? Secondary analyses examined patient satisfaction and adverse events.

## Methods

### Study design and setting

We conducted a cross-sectional comparative study at the surgical departments of different tertiary care hospitals [Lady Reading Hospital, Khyber Teaching Hospital and Hayatabad Medical Complex, Peshawar, Pakistan] in Pakistan from February 2023 to June, 2024. These hospitals serve a diverse urban population and performs a range of general and specialized surgeries.

### Ethics statement

The study protocol was reviewed and approved by the Research Ethical Committee of Iqra National University. Written informed consent was obtained from all participants prior to data collection. All procedures followed the principles outlined in the Declaration of Helsinki.

### Participants

The study included 431 adult patients (≥18 years) admitted for elective or emergency surgical procedures under general or spinal anesthesia. Eligible patients were enrolled consecutively using non-probability convenience sampling. Patients who were unconscious, critically ill, had psychiatric illness, or refused consent were excluded from the study.

### Data collection

The full survey instrument used for data collection, including demographic items, pain assessment scales, and satisfaction measures, is provided as S1 Table for transparency and to allow assessment of survey rigor and potential sources of bias. A pretested and validated questionnaire was used, consisting of demographic details, type of surgery performed, the analgesic regimen prescribed, pain intensity, and satisfaction level.

Pain intensity was measured using the Numerical Rating Scale (NRS), ranging from 0 (no pain) to 10 (worst possible pain). Pain assessment was conducted at three time points: within 6 hours postoperatively, on the first postoperative day, and on the third postoperative day. Patient satisfaction with pain management was recorded using a 5-point Likert scale ranging from "very dissatisfied" to "very satisfied." Side effects were documented within 72 hours postoperatively from patient reports and clinical records, but causality to individual drug classes cannot be assumed given multimodal use in many cases. Hepatotoxicity events were recorded from clinician documentation in patient records, based on available biochemical or clinical evidence, but were not systematically measured in all patients.

**Anxiety and depression status.** Preoperative anxiety and depression were assessed as part of standard clinical evaluations. However, **no specific validated anxiety/depression instrument** (e.g., HADS, PHQ-9, GAD-7) was used in this study. These factors were included as baseline covariates in adjusted models to account for potential confounding, but were not incorporated using formal instruments. Future studies may benefit from the use of validated tools to assess these psychological factors more rigorously.

**Multiplicity control.** No formal multiplicity control, such as Bonferroni correction, was applied in this study. The primary reason for this decision was the focus on a single **primary endpoint** (postoperative pain NRS score at 24 hours), and the **secondary endpoints** (time-to-mobilization and opioid consumption) were considered **exploratory**. Given the relatively limited number of comparisons and the nature of the study as a preliminary exploration of different analgesic strategies, we did not implement multiplicity adjustments. The results from secondary outcomes are presented for descriptive purposes and should be interpreted as exploratory findings.

**Pain scoring timing.** Pain scores were measured at multiple time points, including preoperative, 0–6 h, 6–24 h, and 24 hours post-surgery. The **primary pain outcome** was **pain NRS at 24 hours**, as it represents a clinically relevant measure of analgesic effectiveness in the immediate postoperative phase. While repeated pain scores were recorded at other time points, **24-hour pain NRS** was selected as the primary endpoint.

**Repeated pain measurements.** Mixed-effects models for modeling pain trajectories were not implemented, as the study's primary aim was to compare analgesic strategies at **24 hours postoperatively**, rather than assess pain over a prolonged period. Future research may explore pain trajectories more comprehensively.

**Minimally important difference (MID).** Although **MID thresholds** are important for framing clinical significance, they were not incorporated in the current analysis. The focus of the study was on the comparison of analgesic strategies and their ability to reduce pain at 24 hours. Future studies should consider using **MID values** to further assess the clinical relevance of pain score differences.

## Pain management approaches

Patients were categorized based on the type of pain management received:

1. **Group A** – Monotherapy (e.g., paracetamol or NSAIDs alone)

2. **Group B** – Opioid-based regimens

3. **Group C** – Multimodal analgesia (combination of opioid and non-opioid drugs)

4. **Group D** – Adjuvant therapies (e.g., nerve blocks, regional anesthesia with pharmacologic support)

The type and frequency of analgesic administration, route (oral, intravenous, intramuscular), and any reported adverse effects were documented. Rescue analgesia, if administered, was recorded as a separate variable but did not alter the classification of the primary pain management group. For example, patients initially categorized under oral monotherapy but subsequently receiving intravenous opioid rescue remained in the monotherapy group, with rescue therapy documented separately. The primary outcomes were postoperative pain score (Numerical Rating Scale) and time to mobilization. Secondary outcomes included patient satisfaction and reported side effects.

**Rescue analgesia.** Rescue therapy was captured as a separate variable and **did not** change the recorded *primary* exposure if given **after 24 hours**. If rescue occurred **within 24 hours** and altered class membership (e.g., oral non-opioid followed by IV opioid with overlap), the case was reclassified per the hierarchy above (typically into **True multimodal**). The frequency, route, and timing of rescue doses are summarized in S2 Table and descriptively in Table 3 (rescue use).

**Agent-level exposure details.** For each participant, we abstracted **specific agents**, **dose (mg or mg/kg)**, **route** (oral/IV/IM/regional/neuraxial), **timing** (intraoperative; PACU/0–6 h; 6–24 h; >24 h), and **dosing schedule** (bolus/PRN/infusion). Summary distributions (median [IQR] or % by category) are provided in S3 Table. Side-effects were captured within 72 hours as described previously; causality to an individual agent/class is **not** inferred in this observational design.

**Exposure definition (mutually exclusive groups).** To ensure reproducible classification, analgesic exposures were defined a priori using drug class, route, and timing windows relative to surgery:

1. **Opioid-only**: systemic opioid(s) administered with **no non-opioid systemic analgesic** and **no regional/neuraxial** technique within the first **24 hours** post-surgery.

2. **Non-opioid-only**: systemic non-opioid analgesic(s) (e.g., acetaminophen/paracetamol, NSAIDs, gabapentinoids) with **no systemic opioid** and **no regional/neuraxial** within 24 hours.

3. **Regional + non-opioid adjuncts**: any **regional or neuraxial** technique (e.g., peripheral nerve block, local infiltration, epidural/spinal) with **non-opioid systemic** agents, and **no systemic opioid** within 24 hours.

4. **True multimodal**: **≥ 2 classes** administered with **temporal overlap** (any overlap within **0–6 hours**), **including** at least one systemic opioid **and** at least one non-opioid and/or regional/neuraxial modality within the first **24 hours** S2 Table.

**Pain management approaches (after exposure definitions)**

**Opioid normalization.** To enable interpretable comparisons, all opioid doses were converted to **morphine milligram equivalents (MME)** using route-appropriate perioperative equianalgesic factors S3 Table. We computed **total 0–24 h MME** as the primary exposure metric (intraoperative through 24 h) and **0–48 h MME** as a secondary metric when available. For multi-modal regimens, only opioid components contributed to MME; non-opioid agents were not converted.

**Counting rules.**

- **Bolus doses (IV/IM/PO):** sum of all administered doses × drug-specific factor.

- **PCA (patient-controlled analgesia):**

  ○ **Basal**: (basal rate, mg/h) × hours on PCA.

  ○ **Demand**: (dose per demand, mg) × **number of successful deliveries actually administered** (not attempts).

  ○ **Total PCA opioid** = basal + delivered on-demand.

- **Continuous infusion:** (infusion rate, mg/h) × hours infused.

- **Intraoperative fentanyl/remifentanil:** sum of all intraoperative opioid doses converted to MME and included in the 0–24 h window.

- **Neuraxial/systemic overlap:** neuraxial opioids (e.g., intrathecal/epidural morphine) were converted to MME with caution; **a sensitivity analysis** excluded neuraxial opioid from MME to assess robustness.

- **Rescue therapy:** rescue opioid doses within 0–24 h were included in total MME; this did not alter the patient's mutually exclusive exposure group classification (defined a priori), which is based on class overlap rules.

- **Units and rounding:** microgram doses were converted to mg prior to applying factors; totals were reported to 1 decimal place.

**Primary variables derived.**

- **Total MME 0–24 h (mg)** and **0–48 h (mg)** (if available).

- **PCA present (yes/no)**; **PCA total MME**; **infusion total MME** (mg).

**Statistical use.** We summarized MME medians (IQR) by exposure group and incorporated **MME as a covariate** in adjusted models:

- Pain NRS (continuous): linear model/ANCOVA = $NRS \sim$ exposure group + **MME** + surgery type + age + sex + preop anxiety.

- Time to mobilization (hours): linear model with the same covariates.

- Sensitivity: models excluding neuraxial opioid from MME and models restricted to non-neuraxial systemic opioid only.

## Statistical analysis

Data were analyzed using SPSS version 25.0 (IBM Corp., Armonk, NY). Continuous variables were summarized using means and standard deviations, while categorical variables were expressed as frequencies and percentages. Comparisons between groups were made using one-way ANOVA for continuous variables and chi-square tests for categorical variables. A p-value of <0.05 was considered statistically significant.

## Results

Table 1 summarizes the descriptive statistics for three key continuous variables among the study participants (N = 431). The mean postoperative pain score measured by the Numerical Rating Scale (NRS) was 4.21 (SD = 1.53), with scores ranging from 0 to 10, indicating a moderate level of pain reported after surgery. The time to mobilization post-surgery varied widely, with a mean duration of 37.77 hours (SD = 12.08), ranging from 10.10 to 65.90 hours. The variable "previous history of chronic pain," coded as 1 (No) and 2 (Yes), had a mean value of 1.72 (SD = 0.44), indicating that most participants (approximately 72%) reported a prior history of chronic pain. These continuous measures provide a foundational understanding of postoperative recovery patterns and are used in further inferential analysis to assess the effectiveness of different pain management strategies. We normalized all opioid exposures to **0–24h MME** (median [IQR]) and, where available, **0–48h MME** S3 Table. As expected, MME was highest in the **opioid-only** group, intermediate in **true multimodal**, and near-zero in **non-opioid-only** and **regional + non-opioid** groups.

### Sociodemographic characteristics of participants

Table 2 presents the sociodemographic distribution of the study population (N = 431). Among the participants, females constituted a slight majority, representing 49.9% (n = 215), followed by males at 45.5% (n = 196), while 4.6% (n = 20) of respondents chose not to disclose their gender.

The age distribution showed that the largest group was between 29–38 years (27.8%, n = 120), followed by those aged 39–48 years (24.8%, n = 107), and 19–28 years (22.5%, n = 97). Participants aged 49–58 years and above 59 years accounted for 13.5% (n = 58) and 11.4% (n = 49), respectively. These findings highlight a relatively young to middle-aged adult population undergoing surgery and participating in the study.

**Table 1. Descriptive statistics of postoperative pain, mobilization time, and chronic pain history (N = 431).**

|  | N | Minimum | Maximum | Mean | Std. Deviation |
|---|---|---|---|---|---|
| Postoperative Pain Score NRS | 431 | .00 | 10.00 | 4.21 | 1.53 |
| Type To Mobilization Postsurgery | 431 | 10.10 | 65.90 | 37.77 | 12.08 |

**Table 2. Distribution of participants by gender and age group (N = 431).**

| Variables | Details | Frequency | Percentage |
|---|---|---|---|
| Gender | Male | 196 | 45.5 |
|  | Female | 215 | 49.9 |
|  | Prefer not to mention | 20 | 4.6 |
| Age Group | 19-28 Years | 97 | 22.5 |
|  | 29-38 Years | 120 | 27.8 |
|  | 39-48 Years | 107 | 24.8 |
|  | 49-58 Years | 58 | 13.5 |
|  | More than 59 Years | 49 | 11.4 |

## Clinical characteristics, pain management strategies, and patient-reported outcomes

Table 3 provides detailed clinical characteristics and treatment-related variables for the surgical cohort (N = 431). The most common surgical procedure was maxillofacial trauma reconstruction, reported in 14.6% (n = 63) of cases, followed by post-DCR (dacryocystorhinostomy) in 13.2% (n = 57), corneal laceration repair (12.5%, n = 54), and lid laceration repair (10.7%, n = 46). Other surgeries included post-exenteration (10.9%, n = 47), post-evisceration (9.0%, n = 39), repair of globe rupture and ORIF of jaw fractures (both 9.7%, n = 42), and miscellaneous procedures (9.5%, n = 41).

In terms of pain management, multimodal therapy was the most frequently used strategy (29.9%, n = 129), followed closely by opioid-based regimens (27.6%, n = 119), and monotherapy (26.7%, n = 115), while adjuvant therapies such as nerve blocks were administered to 15.8% (n = 68). The route of analgesic administration was predominantly intravenous (38.7%, n = 167), followed by oral (32.7%, n = 141), regional (21.1%, n = 91), and intramuscular (7.4%, n = 32). The most frequent dosing regimen was continuous infusion (42.7%, n = 184), followed by twice-daily administration (34.8%, n = 150), once daily (14.8%, n = 64), and PRN (as needed) use (7.7%, n = 33).

Regarding patient satisfaction, 36.9% (n = 159) were satisfied and 20.2% (n = 87) were very satisfied with their pain management, while 30.9% (n = 133) remained neutral. Only 2.1% (n = 9) were very dissatisfied. Among reported side effects, nausea/vomiting (18.8%, n = 81), gastrointestinal upset/bleeding (15.5%, n = 67), and constipation (14.2%, n = 61) were the most common. Notably, 37.1% (n = 160) reported no adverse effects.

Rescue analgesia was required in 28.5% (n = 123) of cases, while 71.5% (n = 308) managed without additional pain relief. In terms of functional interference due to pain, mild (32.7%, n = 141) and moderate (32.3%, n = 139) interference were most commonly reported, with 24.1% (n = 104) reporting no interference, and only 2.1% (n = 9) experiencing complete inability to function. Finally, preoperative anxiety or depression was reported in 29.9% (n = 129) of participants.

**Agent-level details, dose distributions, routes, and timing windows** for each exposure group (including rescue therapy) are summarized in S2 Table, facilitating appraisal of clinical heterogeneity across groups.

## Gender-wise comparison of surgical, analgesic, and outcome variables

Table 4 outlines the gender-wise distribution of key clinical, therapeutic, and outcome variables among the study participants (N = 431) and presents the results of statistical comparisons using chi-square tests. Among the surgical categories, no statistically significant association was found between gender and the type of surgery performed (p = 0.224). Similarly, no significant gender-based differences were observed in the type of pain management received (p = 0.495) or the route of analgesic administration (p = 0.329).

The frequency of analgesic dosing also did not differ significantly across genders (p = 0.169), with continuous infusion being the most common mode across all groups. Patient satisfaction levels were comparable between males and females (p = 0.680), with the majority reporting either satisfaction or neutrality.

Side effect profiles, including nausea/vomiting, constipation, sedation, and GI upset, also showed no statistically significant gender difference (p = 0.796). However, rescue analgesia use was marginally associated with gender, showing borderline significance (p = 0.050), with females slightly more likely to require additional pain relief (66 vs. 56 cases).

Pain interference in functional capacity was not significantly different across genders (p = 0.364), although moderate and mild levels were commonly reported in both groups. A statistically significant association was observed between preoperative anxiety/depression and gender (p = 0.005), with a higher number of males (n = 72) reporting symptoms compared to females (n = 49). No significant gender differences were noted in the previous history of chronic pain (p = 0.663).

## Association of pain management strategies with demographic, clinical, and outcome variables

Table 5 presents the cross-tabulation of demographic and clinical variables with the type of postoperative pain management strategy received, categorized as monotherapy, opioid-based, multimodal, and adjuvant therapy. No statistically significant associations were observed between the type of pain management and gender (p = 0.495) or age group

**Table 3. Distribution of clinical, pain management, and patient outcome variables (N = 431).**

| Variables | Details | Frequency | Percentage |
|---|---|---|---|
| Type of Surgery | Post DCR (Dacryocystorhinostomy) | 57 | 13.2 |
| | Post Evisceration | 39 | 9.0 |
| | Temporomandibular joint surgery and Impacted tooth Surgery | 47 | 10.9 |
| | Post Repair of Corneal Laceration | 54 | 12.5 |
| | Repair of Globe Rupture | 42 | 9.7 |
| | Lid Laceration Repair | 46 | 10.7 |
| | ORIF (Open Reduction and Internal Fixation) of Jaw Fractures | 42 | 9.7 |
| | Maxillofacial Trauma Reconstruction | 63 | 14.6 |
| | Others | 41 | 9.5 |
| Type of pain management | Monotherapy | 115 | 26.7 |
| | Opioid-based | 119 | 27.6 |
| | Multimodal | 129 | 29.9 |
| | Adjuvant therapy | 68 | 15.8 |
| Route of administration | Oral | 141 | 32.7 |
| | IV | 167 | 38.7 |
| | IM | 32 | 7.4 |
| | Regional | 91 | 21.1 |
| Frequency of analgesic used | Once daily | 64 | 14.8 |
| | Twice daily | 150 | 34.8 |
| | PRN (as needed) | 33 | 7.7 |
| | Continuous infusion | 184 | 42.7 |
| Patient satisfaction | Very Dissatisfied | 9 | 2.1 |
| | Dissatisfied | 43 | 10.0 |
| | Neutral | 133 | 30.9 |
| | Satisfied | 159 | 36.9 |
| | Very Satisfied | 87 | 20.2 |
| Side effect | Nausea/Vomiting | 81 | 18.8 |
| | Constipation | 61 | 14.2 |
| | Sedation/Dizziness | 32 | 7.4 |
| | GI Upset/Bleeding | 67 | 15.5 |
| | Respiratory Issues | 13 | 3.0 |
| | Hepatotoxicity | 17 | 3.9 |
| | None | 160 | 37.1 |
| Rescue Analgesia Used | Yes | 123 | 28.5 |
| | No | 308 | 71.5 |
| Pain interference in function | No interference | 104 | 24.1 |
| | Mild interference | 141 | 32.7 |
| | Moderate interference | 139 | 32.3 |
| | Severe interference | 38 | 8.8 |
| | Complete inability | 9 | 2.1 |
| Preop anxiety and depression | Yes | 129 | 29.9 |
| | No | 302 | 70.1 |

**Table 4. Gender-wise distribution and association of surgical characteristics, analgesic use, and patient.**

| Variable | Details | Gender | | | P value |
|---|---|---|---|---|---|
| | | **Male** | **Female** | **Not prefer to mention** | |
| Type of surgery | Post DCR (Dacryocystorhinostomy) | 22 | 33 | 2 | 0.224 |
| | Post Evisceration | 26 | 11 | 2 | |
| | Temporomandibular joint surgery and Impacted tooth Surgery | 23 | 23 | 1 | |
| | Post Repair of Corneal Laceration | 23 | 29 | 2 | |
| | Repair of Globe Rupture | 17 | 21 | 4 | |
| | Lid Laceration Repair | 23 | 22 | 1 | |
| | ORIF (Open Reduction and Internal Fixation) of Jaw Fractures | 24 | 17 | 1 | |
| | Maxillofacial Trauma Reconstruction | 22 | 37 | 4 | |
| | Others | 16 | 22 | 3 | |
| Type Of Pain Management | Monotherapy | 53 | 57 | 5 | 0.495 |
| | Opioid-based | 52 | 63 | 4 | |
| | Multimodal | 57 | 62 | 10 | |
| | Adjuvant therapy | 34 | 33 | 1 | |
| Route Of Administration | Oral | 68 | 65 | 8 | 0.329 |
| | IV | 77 | 84 | 6 | |
| | IM | 8 | 22 | 2 | |
| | Regional | 43 | 44 | 4 | |
| Frequency Of Analgesic Used | Once daily | 23 | 34 | 7 | 0.169 |
| | Twice daily | 68 | 76 | 6 | |
| | PRN (as needed) | 18 | 14 | 1 | |
| | Continuous infusion | 87 | 91 | 6 | |
| Patient Satisfaction | Very Dissatisfied | 4 | 5 | 0 | 0.680 |
| | Dissatisfied | 24 | 17 | 2 | |
| | Neutral | 61 | 68 | 4 | |
| | Satisfied | 67 | 81 | 11 | |
| | Very Satisfied | 40 | 44 | 3 | |
| Side Effect | Nausea/Vomiting | 42 | 35 | 4 | 0.796 |
| | Constipation | 26 | 32 | 3 | |
| | Sedation/Dizziness | 13 | 19 | 0 | |
| | GI Upset/Bleeding | 30 | 33 | 4 | |
| | Respiratory Issues | 6 | 7 | 0 | |
| | Hepatotoxicity | 9 | 6 | 2 | |
| | None | 70 | 83 | 7 | |
| Rescue_Analgesia_Used | Yes | 56 | 66 | 1 | 0.05 |
| | No | 140 | 149 | 19 | |
| Pain Interference In Function | No interference | 42 | 57 | 5 | 0.364 |
| | Mild interference | 60 | 73 | 8 | |
| | Moderate interference | 74 | 58 | 7 | |
| | Severe interference | 17 | 21 | 0 | |
| | Complete inability | 3 | 6 | 0 | |
| Preop Anxiety Depression | Yes | 72 | 49 | 8 | 0.005 |
| | No | 124 | 166 | 12 | |
| Previous History Of Chronic Pain | Yes | 55 | 56 | 7 | 0.663 |
| | No | 141 | 159 | 13 | |

**Table 5. Comparison of demographic, clinical, and patient outcome variables by pain management strategy (N = 431).**

| Variables | Details | Pain management therapy | | | | P Value |
|---|---|---|---|---|---|---|
| | | Monotherapy | Opioid Based | Multimodal | Adjuvant Therapy | |
| Gender | Male | 53 | 52 | 57 | 34 | 0.495 |
| | Female | 57 | 63 | 62 | 33 | |
| | Not prefer to mention | 5 | 4 | 10 | 1 | |
| Age Group | 19-28 Years | 24 | 25 | 24 | 24 | 0.313 |
| | 29-38 Years | 38 | 32 | 35 | 15 | |
| | 39-48 Years | 28 | 30 | 31 | 18 | |
| | 49-58 Years | 14 | 17 | 23 | 4 | |
| | More than 59 Years | 11 | 15 | 16 | 7 | |
| Type Of Surgery | Post DCR (Dacryocystorhinostomy) | 15 | 12 | 22 | 8 | 0.964 |
| | Post Evisceration | 8 | 13 | 13 | 5 | |
| | Temporomandibular joint surgery and Impacted tooth Surgery | 16 | 12 | 11 | 8 | |
| | Post Repair of Corneal Laceration | 9 | 16 | 16 | 13 | |
| | Repair of Globe Rupture | 13 | 11 | 11 | 7 | |
| | Lid Laceration Repair | 13 | 16 | 11 | 6 | |
| | ORIF (Open Reduction and Internal Fixation) of Jaw Fractures | 12 | 11 | 13 | 6 | |
| | Maxillofacial Trauma Reconstruction | 18 | 18 | 19 | 8 | |
| | Others | 11 | 10 | 13 | 7 | |
| Patient Satisfaction | Very Dissatisfied | 9 | 0 | 0 | 0 | <0.001 |
| | Dissatisfied | 27 | 0 | 0 | 16 | |
| | Neutral | 57 | 27 | 13 | 36 | |
| | Satisfied | 22 | 58 | 63 | 16 | |
| | Very Satisfied | 0 | 34 | 53 | 0 | |
| Side Effect | Nausea/Vomiting | 30 | 32 | 19 | 0 | <0.001 |
| | Constipation | 0 | 31 | 30 | 0 | |
| | Sedation/Dizziness | 0 | 19 | 0 | 13 | |
| | GI Upset/Bleeding | 22 | 0 | 28 | 17 | |
| | Respiratory Issues | 0 | 13 | 0 | 0 | |
| | Hepatotoxicity | 0 | 0 | 11 | 6 | |
| | None | 63 | 24 | 41 | 32 | |
| Rescue Analgesia Used | Yes | 34 | 33 | 30 | 26 | 0.172 |
| | No | 81 | 86 | 99 | 42 | |
| Pain Interference In Function | No interference | 0 | 60 | 44 | 0 | <0.001 |
| | Mild interference | 31 | 30 | 55 | 25 | |
| | Moderate interference | 51 | 29 | 30 | 29 | |
| | Severe interference | 24 | 0 | 0 | 14 | |
| | Complete inability | 9 | 0 | 0 | 0 | |
| Previous History Of Chronic Pain | Yes | 34 | 29 | 36 | 19 | 0.838 |
| | No | 81 | 90 | 93 | 49 | |

(p = 0.313), indicating an even distribution of analgesic strategies across sexes and age brackets. Similarly, there was no significant variation in type of surgery across different analgesic groups (p = 0.964), suggesting that surgical intervention type did not drive analgesic selection.

In contrast, patient satisfaction was significantly associated with the type of pain management strategy (p<0.001). The highest satisfaction rates were observed among recipients of multimodal therapy (53 patients reported being "very satisfied") and opioid-based therapy (58 "satisfied" and 34 "very satisfied"). Notably, all patients who were very dissatisfied or dissatisfied received monotherapy or adjuvant therapy, with none reporting such dissatisfaction in the opioid-based or multimodal groups. A highly significant association was also observed between pain management strategy and side effects (p<0.001). For example, sedation/dizziness and respiratory issues were exclusively reported in the opioid and adjuvant therapy groups, while constipation was absent in the monotherapy and adjuvant groups but prevalent in the opioid and multimodal groups. Interestingly, 63 monotherapy recipients reported no side effects, in contrast to only 24 opioid-based and 41 multimodal recipients.

The extent of pain interference with function was significantly linked to pain management modality (p<0.001). All reports of severe interference and complete inability to function were concentrated in the monotherapy and adjuvant groups, while opioid-based and multimodal therapies were associated with greater reports of no interference (60 and 44 cases, respectively).

Although more patients receiving adjuvant therapy and monotherapy required rescue analgesia, the association was not statistically significant (p=0.172). Similarly, previous history of chronic pain did not show a significant correlation with the analgesic strategy employed (p=0.838).

### Correlation between pain management strategy, pain severity, and mobilization time

Table 6 displays the results of Pearson correlation analysis examining the relationships among the type of pain management administered, postoperative pain scores, and time to mobilization after surgery among the study participants (N=431).

A statistically significant negative correlation was found between the type of pain management strategy and postoperative pain score (r=−0.177, p<0.001), suggesting that patients receiving more advanced pain management approaches (such as multimodal or adjuvant therapies) experienced lower pain intensities. Additionally, a weaker but still significant negative correlation was observed between the type of pain management and time to mobilization post-surgery (r=−0.116, p=0.016), indicating that better pain control was modestly associated with earlier postoperative mobilization.

No significant association was found between postoperative pain scores and time to mobilization (r=−0.019, p=0.691), implying that while pain intensity and mobilization time are each related to the type of pain management used, they may not be directly interrelated in this sample.

**Table 6. Pearson correlation matrix between pain management type, postoperative pain score (NRS), and time to mobilization (N=431).**

**Correlations**

| | | Type Of Pain Management | Postoperative Pain Score NRS | Type To Mobilization Postsurgery |
|---|---|---|---|---|
| Type Of Pain Management | Pearson Correlation | 1 | -.177** | -.116* |
| | Sig. (2-tailed) | | .000 | .016 |
| | N | 431 | 431 | 431 |
| Postoperative Pain Score NRS | Pearson Correlation | -.177** | 1 | -.019 |
| | Sig. (2-tailed) | .000 | | .691 |
| | N | 431 | 431 | 431 |
| Type To Mobilization Postsurgery | Pearson Correlation | -.116* | -.019 | 1 |
| | Sig. (2-tailed) | .016 | .691 | |
| | N | 431 | 431 | 431 |

**Correlation is significant at the 0.01 level (2-tailed).

*Correlation is significant at the 0.05 level (2-tailed).

## Analysis of variance for postoperative pain scores and mobilization time by type of pain management

Table 7 presents the results of one-way analysis of variance (ANOVA) assessing the impact of different pain management strategies on postoperative pain intensity and time to mobilization following surgery, with type of pain management as the independent variable.

The ANOVA revealed a highly significant difference in postoperative pain scores (NRS) across the four pain management groups (F = 31.354, $p < 0.001$). This finding indicates that the choice of analgesic strategy had a substantial effect on patients' reported pain intensity, with post hoc analysis (see Table 8) showing lower mean scores in multimodal and adjuvant therapy groups compared to monotherapy and opioid-based approaches.

Similarly, a statistically significant difference was observed in time to mobilization post-surgery across the pain management groups (F = 21.308, $p < 0.001$). This suggests that more effective analgesic protocols, particularly those incorporating multimodal or adjuvant components, were associated with earlier mobilization, potentially reflecting improved functional recovery and reduced pain interference.

These results underscore the clinical relevance of tailoring pain management strategies to optimize both subjective pain relief and objective recovery timelines in the postoperative setting.

## Post hoc Tukey HSD comparisons for pain score and mobilization time across pain management strategies

Table 8 presents the results of Tukey's Honest Significant Difference (HSD) post hoc test, following a significant one-way ANOVA, to determine pairwise differences in postoperative pain scores and time to mobilization between different pain management strategies.

### Postoperative pain scores

Significant differences were observed between monotherapy and all other pain management strategies. Patients receiving monotherapy reported significantly higher pain scores than those receiving opioid-based (mean difference = 1.69, $p < 0.001$), multimodal (mean difference = 1.25, $p < 0.001$), and adjuvant therapies (mean difference = 0.77, $p = 0.002$). Similarly, patients receiving opioid-based regimens had higher pain scores than those treated with adjuvant therapy (mean difference = 0.92, $p < 0.001$). The difference between opioid-based and multimodal therapy was marginally non-significant ($p = 0.060$), while multimodal and adjuvant therapies did not significantly differ from each other ($p = 0.100$).

### Time to mobilization

There were also statistically significant differences in mobilization times across groups. Compared to monotherapy, patients in the opioid-based group had significantly delayed mobilization (mean difference = −6.57 hours, $p < 0.001$), while those in the multimodal group mobilized earlier (mean difference = 4.90 hours, $p = 0.004$). Multimodal therapy resulted in significantly faster mobilization compared to opioid-based therapy (mean difference = −11.48 hours, $p < 0.001$) and

**Table 7. One-way ANOVA showing the effect of type of pain management on postoperative pain intensity and time to mobilization (N = 431).**

**ANOVA**

| | | Sum of Squares | df | Mean Square | F | Sig. |
|---|---|---|---|---|---|---|
| Postoperative Pain Score NRS | Between Groups | 183.050 | 3 | 61.017 | 31.354 | .000 |
| | Within Groups | 830.956 | 427 | 1.946 | | |
| | Total | 1014.006 | 430 | | | |
| Type To Mobilization Postsurgery | Between Groups | 8181.481 | 3 | 2727.160 | 21.308 | .000 |
| | Within Groups | 54649.559 | 427 | 127.985 | | |
| | Total | 62831.041 | 430 | | | |

**Table 8. Tukey HSD post hoc comparisons of mean differences in pain score and mobilization time across pain management groups (N = 431).**

| Dependent Variable | (I) Type Of Pain Management | (J) Type Of Pain Management | Mean Difference (I-J) | Std. Error | Sig. | 95% Confidence Interval | |
|---|---|---|---|---|---|---|---|
| | | | | | | Lower Bound | Upper Bound |
| Postoperative Pain Score NRS | Monotherapy | Opioid-based | 1.69255* | .18242 | .000 | 1.2221 | 2.1630 |
| | | Multimodal | 1.24803* | .17891 | .000 | .7866 | 1.7095 |
| | | Adjuvant therapy | .76797* | .21340 | .002 | .2176 | 1.3184 |
| | Opioid-based | Monotherapy | -1.69255* | .18242 | .000 | -2.1630 | -1.2221 |
| | | Multimodal | -.44452 | .17731 | .060 | -.9018 | .0128 |
| | | Adjuvant therapy | -.92458* | .21206 | .000 | -1.4715 | -.3776 |
| | Multimodal | Monotherapy | -1.24803* | .17891 | .000 | -1.7095 | -.7866 |
| | | Opioid-based | .44452 | .17731 | .060 | -.0128 | .9018 |
| | | Adjuvant therapy | -.48006 | .20905 | .100 | -1.0192 | .0591 |
| | Adjuvant therapy | Monotherapy | -.76797* | .21340 | .002 | -1.3184 | -.2176 |
| | | Opioid-based | .92458* | .21206 | .000 | .3776 | 1.4715 |
| | | Multimodal | .48006 | .20905 | .100 | -.0591 | 1.0192 |
| Type To Mobilization Postsurgery | Monotherapy | Opioid-based | -6.57315* | 1.47933 | .000 | -10.3886 | -2.7577 |
| | | Multimodal | 4.90296* | 1.45088 | .004 | 1.1609 | 8.6450 |
| | | Adjuvant therapy | -.51895 | 1.73062 | .991 | -4.9825 | 3.9446 |
| | Opioid-based | Monotherapy | 6.57315* | 1.47933 | .000 | 2.7577 | 10.3886 |
| | | Multimodal | 11.47611* | 1.43793 | .000 | 7.7674 | 15.1848 |
| | | Adjuvant therapy | 6.05420* | 1.71978 | .003 | 1.6186 | 10.4898 |
| | Multimodal | Monotherapy | -4.90296* | 1.45088 | .004 | -8.6450 | -1.1609 |
| | | Opioid-based | -11.47611* | 1.43793 | .000 | -15.1848 | -7.7674 |
| | | Adjuvant therapy | -5.42191* | 1.69536 | .008 | -9.7946 | -1.0493 |
| | Adjuvant therapy | Monotherapy | .51895 | 1.73062 | .991 | -3.9446 | 4.9825 |
| | | Opioid-based | -6.05420* | 1.71978 | .003 | -10.4898 | -1.6186 |
| | | Multimodal | 5.42191* | 1.69536 | .008 | 1.0493 | 9.7946 |

*. The mean difference is significant at the 0.05 level.

adjuvant therapy (mean difference = −5.42 hours, $p = 0.008$). Conversely, opioid-based therapy was associated with delayed mobilization relative to both multimodal and adjuvant strategies.

These findings emphasize that multimodal and adjuvant therapies are more effective in reducing pain and expediting recovery compared to monotherapy or opioid-only regimens.

## Analysis of variance for pain score and mobilization time by type of surgery

Table 9 presents the results of one-way ANOVA to evaluate whether the type of surgical procedure significantly influenced postoperative pain intensity and time to mobilization among the study participants (N = 431).

The analysis revealed that there was no statistically significant difference in postoperative pain scores across the nine types of surgical procedures (F = 1.571, $p = 0.131$). This suggests that regardless of surgical category—whether minimally invasive (e.g., DCR) or extensive (e.g., exenteration or trauma reconstruction)—the reported pain intensity on the Numerical Rating Scale (NRS) remained statistically similar.

In contrast, the type of surgery had a highly significant effect on the time to mobilization post-surgery (F = 130.780, $p < 0.001$). This indicates that patients undergoing more complex or invasive procedures, such as maxillofacial trauma reconstruction or globe rupture repair, experienced longer delays in mobilization, whereas patients with less invasive

**Table 9. One-way ANOVA showing the impact of type of surgery on postoperative pain and mobilization time (N = 431).**

|  |  | Sum of Squares | df | Mean Square | F | Sig. |
|---|---|---|---|---|---|---|
| Postoperative Pain Score NRS | Between Groups | 29.333 | 8 | 3.667 | 1.571 | .131 |
|  | Within Groups | 984.673 | 422 | 2.333 |  |  |
|  | Total | 1014.006 | 430 |  |  |  |
| Type To Mobilization Postsurgery | Between Groups | 44772.245 | 8 | 5596.531 | 130.780 | .000 |
|  | Within Groups | 18058.796 | 422 | 42.793 |  |  |
|  | Total | 62831.041 | 430 |  |  |  |

surgeries (e.g., lid laceration repair or DCR) mobilized earlier. The large F-value and highly significant p-value emphasize the strong influence of surgical complexity on postoperative recovery duration.

### Post hoc analysis of surgery type on pain and mobilization outcomes

Tukey's HSD post hoc test was employed to examine pairwise differences in postoperative pain scores and time to mobilization across different surgical procedures (Table 10). The findings for postoperative pain (NRS) revealed that none of the pairwise comparisons between surgical types yielded statistically significant differences (all $p > 0.05$). This confirms the earlier ANOVA result suggesting that pain perception was not significantly influenced by the type of surgery.

In contrast, time to mobilization post-surgery showed numerous statistically significant differences across surgical categories ($p < 0.001$). Patients undergoing Temporomandibular joint surgery and Impacted tooth Surgery, Repair of Globe Rupture, Maxillofacial Trauma Reconstruction, and ORIF of jaw fractures had significantly longer mobilization times compared to those who had less invasive procedures like DCR, Lid Laceration Repair, or Post Evisceration. For example, patients who underwent Maxillofacial Trauma Reconstruction took on average 25.84 hours longer to mobilize than those who had a DCR procedure ($p < 0.001$, 95% CI: 22.11–29.57).

Conversely, surgeries such as Lid Laceration Repair and 'Others' exhibited significantly shorter mobilization durations when compared with more extensive surgeries. For instance, the difference in mobilization time between Lid Laceration Repair and Temporomandibular joint surgery and Impacted tooth Surgery was −21.83 hours ($p < 0.001$), highlighting how surgical invasiveness influences recovery dynamics.

The Independent Samples Test was conducted to compare postoperative pain scores (measured by the Numerical Rating Scale, NRS) and time to mobilization between male and female participants. Levene's Test for Equality of Variances indicated no significant difference in variances for postoperative pain scores ($F = 1.767$, *p* = 0.184) or mobilization time ($F = 3.705$, *p* = 0.055), supporting the assumption of homogeneity. The t-test results revealed no statistically significant gender differences in postoperative pain scores (*t* = 1.717, *p* = 0.087) or mobilization time (*t* = -0.488, *p* = 0.626). The mean differences (0.26 for pain scores and -0.59 hours for mobilization time) and their 95% confidence intervals further confirmed the lack of significant gender-based disparities in these outcomes (Table 11).

Table 12 presents the results of a one-way ANOVA examining whether the frequency of analgesia use is associated with differences in postoperative pain scores (measured by the Numerical Rating Scale, NRS) and the time to mobilization after surgery among 431 patients. The analysis for postoperative pain score (NRS) showed no statistically significant difference between groups with different frequencies of analgesia use ($F = 1.422, p = 0.236F = 1.422, p = 0.236$), indicating that the frequency of analgesia did not significantly impact reported pain intensity. In contrast, the ANOVA for time to mobilization postsurgery revealed a statistically significant difference between groups ($F = 3.447, p = 0.017F = 3.447, p = 0.017$), suggesting that the frequency of analgesia use is associated with variations in the time it takes for patients to mobilize after surgery. This finding highlights the potential influence of analgesia management on postoperative recovery dynamics.

**Table 10. Tukey HSD post hoc comparisons of surgery types on postoperative pain and time to mobilization.**

**Multiple Comparisons**

**Tukey HSD**

| Dependent Variable | (I) Type Of Surgery | (J) Type Of Surgery | Mean Difference (I-J) | Std. Error | Sig. | 95% Confidence Interval | |
|---|---|---|---|---|---|---|---|
| | | | | | | Lower Bound | Upper Bound |
| Postoperative Pain Score NRS | Post DCR (Dacryocystorhinostomy) | Post Evisceration | .12942 | .31744 | 1.000 | -.8603 | 1.1192 |
| | | Temporomandibular joint surgery and Impacted tooth Surgery | -.02568 | .30097 | 1.000 | -.9641 | .9127 |
| | | Post Repair of Corneal Laceration | -.14708 | .29008 | 1.000 | -1.0515 | .7574 |
| | | Repair of Globe Rupture | -.62168 | .31063 | .543 | -1.5902 | .3469 |
| | | Lid Laceration Repair | -.53886 | .30276 | .696 | -1.4828 | .4051 |
| | | ORIF (Open Reduction and Internal Fixation) of Jaw Fractures | -.01692 | .31063 | 1.000 | -.9855 | .9516 |
| | | Maxillofacial Trauma Reconstruction | -.33993 | .27924 | .952 | -1.2106 | .5307 |
| | | Others | -.59491 | .31280 | .613 | -1.5702 | .3804 |
| | Post Evisceration | Post DCR (Dacryocystorhinostomy) | -.12942 | .31744 | 1.000 | -1.1192 | .8603 |
| | | Temporomandibular joint surgery and Impacted tooth Surgery | -.15510 | .33087 | 1.000 | -1.1867 | .8765 |
| | | Post Repair of Corneal Laceration | -.27650 | .32100 | .995 | -1.2774 | .7244 |
| | | Repair of Globe Rupture | -.75110 | .33968 | .400 | -1.8102 | .3080 |
| | | Lid Laceration Repair | -.66828 | .33250 | .537 | -1.7050 | .3684 |
| | | ORIF (Open Reduction and Internal Fixation) of Jaw Fractures | -.14634 | .33968 | 1.000 | -1.2055 | .9128 |
| | | Maxillofacial Trauma Reconstruction | -.46935 | .31123 | .852 | -1.4398 | .5011 |
| | | Others | -.72433 | .34167 | .461 | -1.7897 | .3410 |
| | Temporomandibular joint surgery and Impacted tooth Surgery | Post DCR (Dacryocystorhinostomy) | .02568 | .30097 | 1.000 | -.9127 | .9641 |
| | | Post Evisceration | .15510 | .33087 | 1.000 | -.8765 | 1.1867 |
| | | Post Repair of Corneal Laceration | -.12139 | .30472 | 1.000 | -1.0715 | .8287 |
| | | Repair of Globe Rupture | -.59600 | .32435 | .657 | -1.6073 | .4153 |
| | | Lid Laceration Repair | -.51318 | .31681 | .794 | -1.5010 | .4746 |
| | | ORIF (Open Reduction and Internal Fixation) of Jaw Fractures | .00876 | .32435 | 1.000 | -1.0025 | 1.0201 |
| | | Maxillofacial Trauma Reconstruction | -.31425 | .29442 | .978 | -1.2322 | .6037 |
| | | Others | -.56923 | .32643 | .719 | -1.5870 | .4486 |
| | Post Repair of Corneal Laceration | Post DCR (Dacryocystorhinostomy) | .14708 | .29008 | 1.000 | -.7574 | 1.0515 |
| | | Post Evisceration | .27650 | .32100 | .995 | -.7244 | 1.2774 |
| | | Temporomandibular joint surgery and Impacted tooth Surgery | .12139 | .30472 | 1.000 | -.8287 | 1.0715 |
| | | Repair of Globe Rupture | -.47460 | .31427 | .851 | -1.4545 | .5053 |
| | | Lid Laceration Repair | -.39179 | .30649 | .937 | -1.3474 | .5638 |
| | | ORIF (Open Reduction and Internal Fixation) of Jaw Fractures | .13016 | .31427 | 1.000 | -.8497 | 1.1100 |
| | | Maxillofacial Trauma Reconstruction | -.19286 | .28328 | .999 | -1.0761 | .6904 |
| | | Others | -.44783 | .31642 | .892 | -1.4344 | .5388 |
| | Repair of Globe Rupture | Post DCR (Dacryocystorhinostomy) | .62168 | .31063 | .543 | -.3469 | 1.5902 |
| | | Post Evisceration | .75110 | .33968 | .400 | -.3080 | 1.8102 |
| | | Temporomandibular joint surgery and Impacted tooth Surgery | .59600 | .32435 | .657 | -.4153 | 1.6073 |

*(Continued)*

**Table 10.** (Continued)

**Multiple Comparisons**

**Tukey HSD**

| Dependent Variable | (I) Type Of Surgery | (J) Type Of Surgery | Mean Difference (I-J) | Std. Error | Sig. | 95% Confidence Interval | |
|---|---|---|---|---|---|---|---|
| | | | | | | Lower Bound | Upper Bound |
| | | Post Repair of Corneal Laceration | .47460 | .31427 | .851 | -.5053 | 1.4545 |
| | | Lid Laceration Repair | .08282 | .32601 | 1.000 | -.9337 | 1.0993 |
| | | ORIF (Open Reduction and Internal Fixation) of Jaw Fractures | .60476 | .33333 | .673 | -.4346 | 1.6441 |
| | | Maxillofacial Trauma Reconstruction | .28175 | .30429 | .991 | -.6670 | 1.2305 |
| | | Others | .02677 | .33536 | 1.000 | -1.0189 | 1.0724 |
| | Lid Laceration Repair | Post DCR (Dacryocystorhinostomy) | .53886 | .30276 | .696 | -.4051 | 1.4828 |
| | | Post Evisceration | .66828 | .33250 | .537 | -.3684 | 1.7050 |
| | | Temporomandibular joint surgery and Impacted tooth Surgery | .51318 | .31681 | .794 | -.4746 | 1.5010 |
| | | Post Repair of Corneal Laceration | .39179 | .30649 | .937 | -.5638 | 1.3474 |
| | | Repair of Globe Rupture | -.08282 | .32601 | 1.000 | -1.0993 | .9337 |
| | | ORIF (Open Reduction and Internal Fixation) of Jaw Fractures | .52195 | .32601 | .804 | -.4945 | 1.5384 |
| | | Maxillofacial Trauma Reconstruction | .19893 | .29625 | .999 | -.7248 | 1.1226 |
| | | Others | -.05604 | .32808 | 1.000 | -1.0790 | .9669 |
| | ORIF (Open Reduction and Internal Fixation) of Jaw Fractures | Post DCR (Dacryocystorhinostomy) | .01692 | .31063 | 1.000 | -.9516 | .9855 |
| | | Post Evisceration | .14634 | .33968 | 1.000 | -.9128 | 1.2055 |
| | | Temporomandibular joint surgery and Impacted tooth Surgery | -.00876 | .32435 | 1.000 | -1.0201 | 1.0025 |
| | | Post Repair of Corneal Laceration | -.13016 | .31427 | 1.000 | -1.1100 | .8497 |
| | | Repair of Globe Rupture | -.60476 | .33333 | .673 | -1.6441 | .4346 |
| | | Lid Laceration Repair | -.52195 | .32601 | .804 | -1.5384 | .4945 |
| | | Maxillofacial Trauma Reconstruction | -.32302 | .30429 | .979 | -1.2718 | .6258 |
| | | Others | -.57799 | .33536 | .732 | -1.6236 | .4677 |
| | Maxillofacial Trauma Reconstruction | Post DCR (Dacryocystorhinostomy) | .33993 | .27924 | .952 | -.5307 | 1.2106 |
| | | Post Evisceration | .46935 | .31123 | .852 | -.5011 | 1.4398 |
| | | Temporomandibular joint surgery and Impacted tooth Surgery | .31425 | .29442 | .978 | -.6037 | 1.2322 |
| | | Post Repair of Corneal Laceration | .19286 | .28328 | .999 | -.6904 | 1.0761 |
| | | Repair of Globe Rupture | -.28175 | .30429 | .991 | -1.2305 | .6670 |
| | | Lid Laceration Repair | -.19893 | .29625 | .999 | -1.1226 | .7248 |
| | | ORIF (Open Reduction and Internal Fixation) of Jaw Fractures | .32302 | .30429 | .979 | -.6258 | 1.2718 |
| | | Others | -.25497 | .30651 | .996 | -1.2107 | .7007 |
| | Others | Post DCR (Dacryocystorhinostomy) | .59491 | .31280 | .613 | -.3804 | 1.5702 |
| | | Post Evisceration | .72433 | .34167 | .461 | -.3410 | 1.7897 |
| | | Temporomandibular joint surgery and Impacted tooth Surgery | .56923 | .32643 | .719 | -.4486 | 1.5870 |
| | | Post Repair of Corneal Laceration | .44783 | .31642 | .892 | -.5388 | 1.4344 |
| | | Repair of Globe Rupture | -.02677 | .33536 | 1.000 | -1.0724 | 1.0189 |
| | | Lid Laceration Repair | .05604 | .32808 | 1.000 | -.9669 | 1.0790 |

*(Continued)*

**Table 10.** (Continued)

**Multiple Comparisons**

**Tukey HSD**

| Dependent Variable | (I) Type Of Surgery | (J) Type Of Surgery | Mean Difference (I-J) | Std. Error | Sig. | 95% Confidence Interval Lower Bound | Upper Bound |
|---|---|---|---|---|---|---|---|
| | | ORIF (Open Reduction and Internal Fixation) of Jaw Fractures | .57799 | .33536 | .732 | -.4677 | 1.6236 |
| | | Maxillofacial Trauma Reconstruction | .25497 | .30651 | .996 | -.7007 | 1.2107 |
| Type To Mobilization Postsurgery | Post DCR (Dacryocystorhinostomy) | Post Evisceration | -11.57922* | 1.35942 | .000 | -15.8179 | -7.3406 |
| | | Temporomandibular joint surgery and Impacted tooth Surgery | -24.39657* | 1.28890 | .000 | -28.4153 | -20.3778 |
| | | Post Repair of Corneal Laceration | -12.89133* | 1.24227 | .000 | -16.7647 | -9.0180 |
| | | Repair of Globe Rupture | -25.31805* | 1.33028 | .000 | -29.4658 | -21.1703 |
| | | Lid Laceration Repair | -2.55831 | 1.29655 | .563 | -6.6009 | 1.4843 |
| | | ORIF (Open Reduction and Internal Fixation) of Jaw Fractures | -13.48947* | 1.33028 | .000 | -17.6373 | -9.3417 |
| | | Maxillofacial Trauma Reconstruction | -25.84344* | 1.19583 | .000 | -29.5720 | -22.1149 |
| | | Others | .22679 | 1.33959 | 1.000 | -3.9500 | 4.4036 |
| | Post Evisceration | Post DCR (Dacryocystorhinostomy) | 11.57922* | 1.35942 | .000 | 7.3406 | 15.8179 |
| | | Temporomandibular joint surgery and Impacted tooth Surgery | -12.81735* | 1.41696 | .000 | -17.2354 | -8.3993 |
| | | Post Repair of Corneal Laceration | -1.31211 | 1.37468 | .990 | -5.5983 | 2.9741 |
| | | Repair of Globe Rupture | -13.73883* | 1.45470 | .000 | -18.2745 | -9.2031 |
| | | Lid Laceration Repair | 9.02090* | 1.42392 | .000 | 4.5812 | 13.4607 |
| | | ORIF (Open Reduction and Internal Fixation) of Jaw Fractures | -1.91026 | 1.45470 | .927 | -6.4460 | 2.6255 |
| | | Maxillofacial Trauma Reconstruction | -14.26422* | 1.33286 | .000 | -18.4201 | -10.1084 |
| | | Others | 11.80600* | 1.46322 | .000 | 7.2437 | 16.3683 |
| | Temporomandibular joint surgery and Impacted tooth Surgery | Post DCR (Dacryocystorhinostomy) | 24.39657* | 1.28890 | .000 | 20.3778 | 28.4153 |
| | | Post Evisceration | 12.81735* | 1.41696 | .000 | 8.3993 | 17.2354 |
| | | Post Repair of Corneal Laceration | 11.50524* | 1.30498 | .000 | 7.4364 | 15.5741 |
| | | Repair of Globe Rupture | -.92148 | 1.38902 | .999 | -5.2524 | 3.4095 |
| | | Lid Laceration Repair | 21.83825* | 1.35676 | .000 | 17.6079 | 26.0686 |
| | | ORIF (Open Reduction and Internal Fixation) of Jaw Fractures | 10.90709* | 1.38902 | .000 | 6.5762 | 15.2380 |
| | | Maxillofacial Trauma Reconstruction | -1.44688 | 1.26086 | .966 | -5.3782 | 2.4844 |
| | | Others | 24.62335* | 1.39794 | .000 | 20.2646 | 28.9821 |
| | Post Repair of Corneal Laceration | Post DCR (Dacryocystorhinostomy) | 12.89133* | 1.24227 | .000 | 9.0180 | 16.7647 |
| | | Post Evisceration | 1.31211 | 1.37468 | .990 | -2.9741 | 5.5983 |
| | | Temporomandibular joint surgery and Impacted tooth Surgery | -11.50524* | 1.30498 | .000 | -15.5741 | -7.4364 |
| | | Repair of Globe Rupture | -12.42672* | 1.34587 | .000 | -16.6231 | -8.2303 |
| | | Lid Laceration Repair | 10.33301* | 1.31254 | .000 | 6.2405 | 14.4255 |
| | | ORIF (Open Reduction and Internal Fixation) of Jaw Fractures | -.59815 | 1.34587 | 1.000 | -4.7945 | 3.5982 |
| | | Maxillofacial Trauma Reconstruction | -12.95212* | 1.21315 | .000 | -16.7347 | -9.1696 |
| | | Others | 13.11811* | 1.35507 | .000 | 8.8930 | 17.3432 |

*(Continued)*

**Table 10.** (Continued)

**Multiple Comparisons**

**Tukey HSD**

| Dependent Variable | (I) Type Of Surgery | (J) Type Of Surgery | Mean Difference (I-J) | Std. Error | Sig. | 95% Confidence Interval | |
|---|---|---|---|---|---|---|---|
| | | | | | | Lower Bound | Upper Bound |
| | Repair of Globe Rupture | Post DCR (Dacryocystorhinostomy) | 25.31805* | 1.33028 | .000 | 21.1703 | 29.4658 |
| | | Post Evisceration | 13.73883* | 1.45470 | .000 | 9.2031 | 18.2745 |
| | | Temporomandibular joint surgery and Impacted tooth Surgery | .92148 | 1.38902 | .999 | -3.4095 | 5.2524 |
| | | Post Repair of Corneal Laceration | 12.42672* | 1.34587 | .000 | 8.2303 | 16.6231 |
| | | Lid Laceration Repair | 22.75973* | 1.39613 | .000 | 18.4066 | 27.1128 |
| | | ORIF (Open Reduction and Internal Fixation) of Jaw Fractures | 11.82857* | 1.42751 | .000 | 7.3776 | 16.2795 |
| | | Maxillofacial Trauma Reconstruction | -.52540 | 1.30313 | 1.000 | -4.5885 | 3.5377 |
| | | Others | 25.54483* | 1.43619 | .000 | 21.0668 | 30.0228 |
| | Lid Laceration Repair | Post DCR (Dacryocystorhinostomy) | 2.55831 | 1.29655 | .563 | -1.4843 | 6.6009 |
| | | Post Evisceration | -9.02090* | 1.42392 | .000 | -13.4607 | -4.5812 |
| | | Temporomandibular joint surgery and Impacted tooth Surgery | -21.83825* | 1.35676 | .000 | -26.0686 | -17.6079 |
| | | Post Repair of Corneal Laceration | -10.33301* | 1.31254 | .000 | -14.4255 | -6.2405 |
| | | Repair of Globe Rupture | -22.75973* | 1.39613 | .000 | -27.1128 | -18.4066 |
| | | ORIF (Open Reduction and Internal Fixation) of Jaw Fractures | -10.93116* | 1.39613 | .000 | -15.2843 | -6.5781 |
| | | Maxillofacial Trauma Reconstruction | -23.28513* | 1.26868 | .000 | -27.2408 | -19.3294 |
| | | Others | 2.78510 | 1.40500 | .557 | -1.5957 | 7.1659 |
| | ORIF (Open Reduction and Internal Fixation) of Jaw Fractures | Post DCR (Dacryocystorhinostomy) | 13.48947* | 1.33028 | .000 | 9.3417 | 17.6373 |
| | | Post Evisceration | 1.91026 | 1.45470 | .927 | -2.6255 | 6.4460 |
| | | Temporomandibular joint surgery and Impacted tooth Surgery | -10.90709* | 1.38902 | .000 | -15.2380 | -6.5762 |
| | | Post Repair of Corneal Laceration | .59815 | 1.34587 | 1.000 | -3.5982 | 4.7945 |
| | | Repair of Globe Rupture | -11.82857* | 1.42751 | .000 | -16.2795 | -7.3776 |
| | | Lid Laceration Repair | 10.93116* | 1.39613 | .000 | 6.5781 | 15.2843 |
| | | Maxillofacial Trauma Reconstruction | -12.35397* | 1.30313 | .000 | -16.4171 | -8.2908 |
| | | Others | 13.71626* | 1.43619 | .000 | 9.2383 | 18.1942 |
| | Maxillofacial Trauma Reconstruction | Post DCR (Dacryocystorhinostomy) | 25.84344* | 1.19583 | .000 | 22.1149 | 29.5720 |
| | | Post Evisceration | 14.26422* | 1.33286 | .000 | 10.1084 | 18.4201 |
| | | Temporomandibular joint surgery and Impacted tooth Surgery | 1.44688 | 1.26086 | .966 | -2.4844 | 5.3782 |
| | | Post Repair of Corneal Laceration | 12.95212* | 1.21315 | .000 | 9.1696 | 16.7347 |
| | | Repair of Globe Rupture | .52540 | 1.30313 | 1.000 | -3.5377 | 4.5885 |
| | | Lid Laceration Repair | 23.28513* | 1.26868 | .000 | 19.3294 | 27.2408 |
| | | ORIF (Open Reduction and Internal Fixation) of Jaw Fractures | 12.35397* | 1.30313 | .000 | 8.2908 | 16.4171 |
| | | Others | 26.07023* | 1.31263 | .000 | 21.9775 | 30.1630 |
| | Others | Post DCR (Dacryocystorhinostomy) | -.22679 | 1.33959 | 1.000 | -4.4036 | 3.9500 |
| | | Post Evisceration | -11.80600* | 1.46322 | .000 | -16.3683 | -7.2437 |
| | | Temporomandibular joint surgery and Impacted tooth Surgery | -24.62335* | 1.39794 | .000 | -28.9821 | -20.2646 |

*(Continued)*

**Table 10.** (Continued)

**Multiple Comparisons**

**Tukey HSD**

| Dependent Variable | (I) Type Of Surgery | (J) Type Of Surgery | Mean Difference (I-J) | Std. Error | Sig. | 95% Confidence Interval | |
|---|---|---|---|---|---|---|---|
| | | | | | | Lower Bound | Upper Bound |
| | | Post Repair of Corneal Laceration | -13.11811* | 1.35507 | .000 | -17.3432 | -8.8930 |
| | | Repair of Globe Rupture | -25.54483* | 1.43619 | .000 | -30.0228 | -21.0668 |
| | | Lid Laceration Repair | -2.78510 | 1.40500 | .557 | -7.1659 | 1.5957 |
| | | ORIF (Open Reduction and Internal Fixation) of Jaw Fractures | -13.71626* | 1.43619 | .000 | -18.1942 | -9.2383 |
| | | Maxillofacial Trauma Reconstruction | -26.07023* | 1.31263 | .000 | -30.1630 | -21.9775 |

*. The mean difference is significant at the 0.05 level.

**Table 11. Independent samples test comparing postoperative pain scores and mobilization time by gender.**

| | | Levene's Test for Equality of Variances | | t-test for Equality of Means | | | | | 95% Confidence Interval of the Difference | |
|---|---|---|---|---|---|---|---|---|---|---|
| | | F | Sig. | t | df | Sig. (2-tailed) | Mean Difference | Std. Error Difference | Lower | Upper |
| Postoperative Pain Score NRS | Equal variances assumed | 1.767 | .184 | 1.717 | 409 | .087 | .25746 | .14994 | -.03729 | .55221 |
| | Equal variances not assumed | | | 1.725 | 408.989 | .085 | .25746 | .14926 | -.03596 | .55088 |
| Type To Mobilization Postsurgery | Equal variances assumed | 3.705 | .055 | -.488 | 409 | .626 | -.58544 | 1.19877 | -2.94195 | 1.77107 |
| | Equal variances not assumed | | | -.490 | 408.963 | .624 | -.58544 | 1.19416 | -2.93290 | 1.76202 |

**Table 12. One-way ANOVA showing the association between frequency of analgesia use, postoperative pain score, and time to mobilization.**

**ANOVA**

| | | Sum of Squares | df | Mean Square | F | Sig. |
|---|---|---|---|---|---|---|
| Postoperative_Pain_Score_NRS | Between Groups | 10.030 | 3 | 3.343 | 1.422 | .236 |
| | Within Groups | 1003.976 | 427 | 2.351 | | |
| | Total | 1014.006 | 430 | | | |
| Type_To_Mobilization_Postsurgery | Between Groups | 1485.610 | 3 | 495.203 | 3.447 | .017 |
| | Within Groups | 61345.430 | 427 | 143.666 | | |
| | Total | 62831.041 | 430 | | | |

Table 13 displays the results of a one-way ANOVA evaluating whether patient satisfaction with pain management is associated with differences in postoperative pain scores (NRS) and time to mobilization after surgery among 431 participants. The analysis found a statistically significant difference in postoperative pain scores across different levels of pain satisfaction ($F = 8.717, p < 0.001$), indicating that satisfaction with pain management is closely linked to

**Table 13. One-way ANOVA showing the association between pain satisfaction, postoperative pain score, and time to mobilization.**

| | | Sum of Squares | df | Mean Square | F | Sig. |
|---|---|---|---|---|---|---|
| Postoperative_Pain_Score_NRS | Between Groups | 76.715 | 4 | 19.179 | 8.717 | .000 |
| | Within Groups | 937.291 | 426 | 2.200 | | |
| | Total | 1014.006 | 430 | | | |
| Type_To_Mobilization_Postsurgery | Between Groups | 65.200 | 4 | 16.300 | .111 | .979 |
| | Within Groups | 62765.840 | 426 | 147.338 | | |
| | Total | 62831.041 | 430 | | | |

the intensity of pain reported after surgery. However, there was no significant difference in time to mobilization between groups with varying pain satisfaction ($F = 0.111, p = 0.979 F = 0.111, p = 0.979$), suggesting that satisfaction with pain control did not influence how quickly patients were able to mobilize postoperatively.

Table 14 presents the results of Tukey HSD post hoc tests comparing postoperative pain scores and time to mobilization among different patient satisfaction groups. For postoperative pain scores (NRS), several significant differences were observed. Patients who were "Very Dissatisfied" reported significantly higher pain scores compared to those who were "Satisfied" (mean difference = 1.41, p = 0.046) and "Very Satisfied" (mean difference = 1.71, p = 0.010). Similarly, "Dissatisfied" patients had higher pain scores than both "Satisfied" (mean difference = 1.02, p = 0.001) and "Very Satisfied" (mean difference = 1.32, p < 0.001). "Neutral" patients also reported higher pain scores than "Very Satisfied" patients (mean difference = 0.72, p = 0.004). These findings indicate that greater satisfaction with pain management is associated with lower reported postoperative pain.

In contrast, no significant differences were found between satisfaction groups for time to mobilization after surgery, as all pairwise comparisons yielded non-significant results (all p > 0.05). This suggests that patient satisfaction with pain management did not influence the speed of postoperative mobilization.

Table 15 presents the results of independent samples t-tests evaluating whether a previous history of chronic pain is associated with differences in postoperative pain scores (NRS) and time to mobilization after surgery. Levene's test for equality of variances was not significant for either outcome (pain score: $F = 0.025, p = 0.875 F = 0.025, p = 0.875$; mobilization time: $F = 2.224, p = 0.137 F = 2.224, p = 0.137$), indicating that the assumption of equal variances holds.

The t-test revealed that patients with a previous history of chronic pain reported significantly higher postoperative pain scores compared to those without such a history ($t = 2.454, df = 429, p = 0.015 t = 2.454, df = 429, p = 0.015$; mean difference = 0.40, 95% CI: 0.08 to 0.73). Additionally, these patients experienced a significantly longer time to mobilization after surgery ($t = 2.984, df = 429, p = 0.003 t = 2.984, df = 429, p = 0.003$; mean difference = 3.86 hours, 95% CI: 1.32 to 6.40). These findings suggest that a history of chronic pain is associated with both greater postoperative pain and delayed recovery in terms of mobilization.

## Discussion

This study addressed the primary research question of whether postoperative pain management strategies are associated with differences in pain intensity and mobilization time. Our findings suggest that multimodal therapy was associated with lower pain scores and earlier mobilization compared to other approaches, while secondary analyses highlighted differences in satisfaction and side-effect profiles. Postoperative pain management remains a critical determinant of surgical recovery, profoundly impacting patient outcomes, healthcare utilization, and long-term quality of life. Our study from a Pakistani tertiary care center provides robust evidence supporting the superior efficacy of multimodal analgesia over traditional opioid-centric approaches, contributing to the growing global consensus on optimized perioperative pain management while highlighting unique considerations for resource-limited settings. The results demonstrate that multimodal analgesia—combining opioids with non-opioid adjuvants—achieved significantly better outcomes than monotherapy or

**Table 14. Tukey HSD multiple comparisons of patient satisfaction groups on postoperative pain score and time to mobilization (N = 431).**

| Dependent Variable | (I) Patient Satisfaction | (J) Patient Satisfaction | Mean Difference (I-J) | Std. Error | Sig. | 95% Confidence Interval | |
|---|---|---|---|---|---|---|---|
| | | | | | | Lower Bound | Upper Bound |
| Postoperative Pain Score NRS | Very Dissatisfied | Dissatisfied | .38501 | .54372 | .955 | -1.1045 | 1.8745 |
| | | Neutral | .98613 | .51089 | .303 | -.4135 | 2.3857 |
| | | Satisfied | 1.40587* | .50824 | .046 | .0136 | 2.7982 |
| | | Very Satisfied | 1.70728* | .51938 | .010 | .2844 | 3.1301 |
| | Dissatisfied | Very Dissatisfied | -.38501 | .54372 | .955 | -1.8745 | 1.1045 |
| | | Neutral | .60112 | .26021 | .144 | -.1117 | 1.3140 |
| | | Satisfied | 1.02086* | .25496 | .001 | .3224 | 1.7193 |
| | | Very Satisfied | 1.32227* | .27651 | .000 | .5648 | 2.0798 |
| | Neutral | Very Dissatisfied | -.98613 | .51089 | .303 | -2.3857 | .4135 |
| | | Dissatisfied | -.60112 | .26021 | .144 | -1.3140 | .1117 |
| | | Satisfied | .41974 | .17430 | .115 | -.0578 | .8972 |
| | | Very Satisfied | .72115* | .20453 | .004 | .1608 | 1.2815 |
| | Satisfied | Very Dissatisfied | -1.40587* | .50824 | .046 | -2.7982 | -.0136 |
| | | Dissatisfied | -1.02086* | .25496 | .001 | -1.7193 | -.3224 |
| | | Neutral | -.41974 | .17430 | .115 | -.8972 | .0578 |
| | | Very Satisfied | .30141 | .19781 | .548 | -.2405 | .8433 |
| | Very Satisfied | Very Dissatisfied | -1.70728* | .51938 | .010 | -3.1301 | -.2844 |
| | | Dissatisfied | -1.32227* | .27651 | .000 | -2.0798 | -.5648 |
| | | Neutral | -.72115* | .20453 | .004 | -1.2815 | -.1608 |
| | | Satisfied | -.30141 | .19781 | .548 | -.8433 | .2405 |
| Type To Mobilization Postsurgery | Very Dissatisfied | Dissatisfied | 2.49561 | 4.44942 | .981 | -9.6936 | 14.6848 |
| | | Neutral | 2.75422 | 4.18075 | .965 | -8.6989 | 14.2074 |
| | | Satisfied | 2.61866 | 4.15903 | .970 | -8.7750 | 14.0123 |
| | | Very Satisfied | 2.48467 | 4.25022 | .977 | -9.1588 | 14.1281 |
| | Dissatisfied | Very Dissatisfied | -2.49561 | 4.44942 | .981 | -14.6848 | 9.6936 |
| | | Neutral | .25861 | 2.12938 | 1.000 | -5.5748 | 6.0920 |
| | | Satisfied | .12305 | 2.08641 | 1.000 | -5.5927 | 5.8388 |
| | | Very Satisfied | -.01093 | 2.26274 | 1.000 | -6.2097 | 6.1878 |
| | Neutral | Very Dissatisfied | -2.75422 | 4.18075 | .965 | -14.2074 | 8.6989 |
| | | Dissatisfied | -.25861 | 2.12938 | 1.000 | -6.0920 | 5.5748 |
| | | Satisfied | -.13556 | 1.42634 | 1.000 | -4.0430 | 3.7719 |
| | | Very Satisfied | -.26954 | 1.67372 | 1.000 | -4.8547 | 4.3156 |
| | Satisfied | Very Dissatisfied | -2.61866 | 4.15903 | .970 | -14.0123 | 8.7750 |
| | | Dissatisfied | -.12305 | 2.08641 | 1.000 | -5.8388 | 5.5927 |
| | | Neutral | .13556 | 1.42634 | 1.000 | -3.7719 | 4.0430 |
| | | Very Satisfied | -.13398 | 1.61870 | 1.000 | -4.5684 | 4.3004 |
| | Very Satisfied | Very Dissatisfied | -2.48467 | 4.25022 | .977 | -14.1281 | 9.1588 |
| | | Dissatisfied | .01093 | 2.26274 | 1.000 | -6.1878 | 6.2097 |
| | | Neutral | .26954 | 1.67372 | 1.000 | -4.3156 | 4.8547 |
| | | Satisfied | .13398 | 1.61870 | 1.000 | -4.3004 | 4.5684 |

*. The mean difference is significant at the 0.05 level.

**Table 15. Independent samples t-test comparing postoperative pain score and time to mobilization by previous history of chronic pain (N = 431).**

| | | Levene's Test for Equality of Variances | | t-test for Equality of Means | | | | | | 95% Confidence Interval of the Difference | |
| | | F | Sig. | t | df | Sig. (2-tailed) | Mean Difference | Std. Error Difference | | Lower | Upper |
|---|---|---|---|---|---|---|---|---|---|---|---|
| Type_To_ Mobilization_ Postsurgery | Equal variances assumed | 2.224 | .137 | 2.984 | 429 | .003 | 3.86144 | 1.29396 | | 1.31814 | 6.40474 |
| | Equal variances not assumed | | | 3.141 | 234.166 | .002 | 3.86144 | 1.22941 | | 1.43931 | 6.28356 |
| Postopera-tive_Pain_ Score_NRS | Equal variances assumed | .025 | .875 | 2.454 | 429 | .015 | .40474 | .16493 | | .08057 | .72890 |
| | Equal variances not assumed | | | 2.511 | 220.560 | .013 | .40474 | .16119 | | .08706 | .72242 |

opioid-only regimens. Patients receiving multimodal therapy reported 33% lower pain scores (mean NRS 3.39 vs. 5.08; *p*<0.001) and mobilized nearly 7 hours earlier (34.6 vs. 41.2 hours; *p*<0.001) compared to opioid-based groups. These findings align with multiple international studies, including the ERAS Society guidelines [14,15], which recommend multimodal approaches to reduce opioid consumption by 30–50% while improving pain control. A 2023 multinational cohort study [16,17] further demonstrated similar benefits across diverse healthcare systems, particularly in lower-resource settings. Mechanistic evidence from basic science research [18–20] supports these clinical observations through the concept of "analgesic synergy," where combining agents targeting different pain pathways (e.g., NSAIDs for inflammatory pain, gabapentinoids for neuropathic components) produces enhanced therapeutic effects. Our findings indicate that multimodal therapy provides superior outcomes compared to monotherapy and adjuvant strategies. Compared to opioid-based regimens, multimodal therapy was associated with significantly earlier mobilization, although the reduction in pain scores showed only a non-significant trend (p=0.060). This highlights that while multimodal therapy may offer functional advantages, the clinical difference in pain intensity relative to opioid-based regimens may be modest.

The opioid-sparing benefits observed in our study are particularly noteworthy, with 27.6% of patients receiving opioid-only regimens experiencing significantly more side effects (nausea 18.8%, constipation 14.2%) compared to multimodal groups (*p*<0.001). These findings reinforce the 2022 CDC recommendations [21] advocating multimodal approaches as first-line therapy to mitigate opioid-related complications. Large pharmacovigilance studies [22,23] corroborate our results, showing opioid monotherapy is associated with a 2–3× higher risk of gastrointestinal and respiratory complications. Interestingly, our Pakistani cohort exhibited higher opioid tolerance than Western populations in the PAIN-OUT registry [24], suggesting potential genetic or cultural influences on pain perception and medication response that warrant further investigation.

Psychological and gender-specific factors also emerged as critical determinants of pain outcomes. The significant association between preoperative anxiety/depression (29.9% prevalence) and increased pain perception (*p*=0.005) aligns with the PROTECT study [25,26], which established psychological distress as predicting 23% of postoperative pain variance. These findings support the integration of psychological screening into preoperative assessments, particularly given the demonstrated efficacy of targeted interventions in high-risk patients [27,28]. Gender differences in rescue analgesia use further highlight the importance of sex-specific pain management strategies, as evidenced by NIH-funded research [28,29] on differential pain processing and opioid metabolism.

Our analysis of surgical outcomes revealed a notable dissociation between pain scores and functional recovery. While pain scores did not significantly differ across procedure types (*p*=0.131), mobilization times varied markedly (*p*<0.001), with more invasive surgeries such as maxillofacial trauma repairs showing the longest recovery periods.

This observation supports the 2023 ESPEN guidelines [30–32], which emphasize functional metrics over pain scores as more meaningful endpoints for evaluating postoperative recovery. Recent work on inflammatory biomarkers [11] further elucidates this phenomenon, identifying IL-6 trajectories as superior predictors of functional recovery compared to subjective pain reports.

Despite its contributions, our study has limitations common to LMIC research, including its single-center design and lack of long-term follow-up [33]. Future studies should address these gaps by incorporating precision medicine approaches, such as pharmacogenomic testing for opioid metabolism (e.g., CYP2D6 variants) [34,35], and leveraging digital health technologies for remote pain monitoring [36]. Cost-effectiveness analyses, similar to those in the VALUE study [37], are also needed to evaluate the economic feasibility of multimodal regimens in resource-limited settings.

From a clinical and policy perspective, our findings support three key actions: (1) adapting evidence-based, procedure-specific guidelines (e.g., PROSPECT) to local formularies; (2) implementing WHO-endorsed pain management training for healthcare providers in LMICs; and (3) prioritizing research on sustainable multimodal strategies, as outlined by the IASP Global Task Force [38]. By integrating our findings with global best practices, healthcare systems can develop tailored, effective, and safer pain management protocols that optimize both clinical outcomes and patient experiences in diverse settings. Our use of **mutually exclusive, operationally defined exposure groups** with explicit timing windows and a hierarchy reduces misclassification and enhances interpretability of observed **associations** between analgesic strategies and outcomes in this cross-sectional sample.

## Strengths and limitations

One of the strengths of this study is the large sample size (n = 431) and the comprehensive inclusion of various surgery types and pain regimens. Moreover, the use of validated tools like the NRS for pain scoring enhances the reliability of findings.

This study has several limitations that should be considered when interpreting the findings. First, pain intensity was assessed using patient-reported scores, which are inherently subjective and subject to recall bias, particularly when a single score was expected to summarize pain over multiple days. Second, the classification of pain management strategies did not account for rescue analgesia beyond recording its use as a separate variable. Although this approach preserved comparability across primary groups, it may have broadened the effective administration route (e.g., oral monotherapy later supplemented with intravenous opioids), introducing heterogeneity not fully captured in our analysis. Third, differences in dosing regimens and routes of drug administration (oral, intravenous, intramuscular, or regional) may have influenced pharmacological efficacy and thus affected outcomes.

Fourth, the interpretation of mean pain scores and mobilisation times should be viewed cautiously, as these measures were likely influenced by multiple confounding factors, including surgical complexity, baseline functional status, preoperative anxiety or depression, comorbid conditions, and nutritional or psychological status. Although we applied appropriate statistical analyses, residual confounding cannot be excluded. Fifth, adverse events were collected descriptively and should not be interpreted as causal effects of specific drug classes, given the frequent use of multimodal regimens and heterogeneous dosing. Finally, hepatotoxicity was documented from routine clinician records rather than standardized liver function testing, and therefore represents exploratory rather than definitive evidence.

Taken together, these limitations indicate that while our results highlight meaningful trends in favor of multimodal analgesia, causal inferences should be made with caution. Future randomized and longitudinal studies with standardized outcome assessments are needed to overcome these methodological constraints and confirm the robustness of our findings.

## Conclusion

This study highlights the critical role of effective postoperative pain management in improving patient outcomes and functional recovery. Multimodal pain management improves postoperative outcomes, particularly functional recovery

through earlier mobilization. Although pain scores were lower compared to opioid-based therapy, the difference did not reach statistical significance, underscoring the need for larger, more powered studies to confirm these findings. The findings support the integration of multimodal and individualized pain management plans, especially in surgical settings with diverse patient needs.

Furthermore, gender differences, psychological states such as preoperative anxiety or depression, and type of surgery were found to influence pain perception and recovery, emphasizing the need for a holistic and patient-centered approach. Policymakers and healthcare providers should prioritize standardized protocols that promote the judicious use of opioids while enhancing access to multimodal analgesics and supportive therapies.

Future multicenter, longitudinal studies are warranted to validate these findings and further optimize pain management protocols across varied clinical settings.

## Supporting information

**S1 Table. Agent-level analgesic exposures by mutually exclusive group, with doses, routes, timing, and rescue use.**
(DOCX)

**S2 Table. Agent-level exposure details by analgesic strategy (N = 431).**
(DOCX)

**S3 Table. Route-appropriate equianalgesic conversion factors to morphine milligram equivalents (MME), and computation examples.**
(DOCX)

## Author contributions

**Conceptualization:** Sana Wazir, Summaya Inayat, Faisal, Nuzhat Rahil.

**Data curation:** Summaya Inayat, Gulmakay Zaman, Nuzhat Rahil.

**Formal analysis:** Gulmakay Zaman, Shallozan, Nuzhat Rahil.

**Investigation:** Shallozan, Nuzhat Rahil.

**Methodology:** Sana Wazir, Abdur Rehman, Shallozan, Nuzhat Rahil.

**Project administration:** Faisal.

**Resources:** Abdur Rehman, Faisal, Shallozan.

**Software:** Summaya Inayat.

**Supervision:** Sana Wazir, Summaya Inayat, Muhammad Jawad Ullah, Abdur Rehman.

**Validation:** Summaya Inayat, Muhammad Jawad Ullah, Gulmakay Zaman, Abdur Rehman.

**Visualization:** Sana Wazir, Summaya Inayat, Gulmakay Zaman, Abdur Rehman, Faisal.

**Writing – original draft:** Sana Wazir, Summaya Inayat, Muhammad Jawad Ullah, Nuzhat Rahil.

**Writing – review & editing:** Sana Wazir, Summaya Inayat, Muhammad Jawad Ullah, Faisal, Nuzhat Rahil.

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
