## [Decision Letter · Decision Letter 0]

16 Sep 2025

PGPH-D-25-01718

Multimodal Analgesia in Resource-Limited Settings: A Comparative Effectiveness Study of Postoperative Pain Management Strategies in Pakistan

Dear Dr. Rahil,

Thank you for submitting your manuscript to PLOS Global Public Health. After careful consideration, we feel that it has merit but does not fully meet PLOS Global Public Health’s publication criteria as it currently stands. Therefore, we invite you to submit a revised version of the manuscript that addresses the points raised during the review process.

We look forward to receiving your revised manuscript.

Kind regards,

Barnabas Tobi Alayande

Academic Editor

Journal Requirements:

2. In the online submission form, you indicated that The data will be available from the corresponding author upon reasonable request.

3. Uploaded as supplementary information.

Reviewers' comments:

Reviewer's Responses to Questions

**Comments to the Author**

1. Does this manuscript meet PLOS Global Public Health’s publication criteria?

Reviewer #1: Partly

Reviewer #2: No

2. Has the statistical analysis been performed appropriately and rigorously?

Reviewer #1: Yes

Reviewer #2: No

3. Have the authors made all data underlying the findings in their manuscript fully available (please refer to the Data Availability Statement at the start of the manuscript PDF file)?

Reviewer #1: Yes

Reviewer #2: No

4. Is the manuscript presented in an intelligible fashion and written in standard English?

Reviewer #1: Yes

Reviewer #2: Yes

Reviewer #1: This is a sound paper addressing a key challenge in post-surgical pain management. There is strong scientific rationale and analytical approaches however some key (5) aspects sneed to be addressed. 1) While multimodal and opioid therapies seem to be the superior pain management classes, there is no evidence to suggest that the difference in the effectiveness (effectiveness on pain management/ outcomes) of the 2 classes is significant. It may be case that one of the drug classes is only slightly more effective. 2) Other key study limitations were not highlighted at a participant and research methodology level e.g. recall bias on pain perception/ true representation of a single score to days of symptoms; pain management choice, despite effectiveness recorded being best for patient physiology, disease pathology, or healing; dosing and method of drug administration affecting optimal drug efficacy; confounding factors to functionality reporting etc. 3) on the methods the questionnaire/ survey administered to participants is not provided making it difficult to determine potential bias/ rigour of survey structure and messaging 4) On the results provided i) the value of the mean pain score/ mobilisation time in light of multiple confounding factors is unclear; ii) it is unclear if pain rescue administration broadens the administration route/ if it is accounted for in analysis(e.g. if mono-therapy oral initially then rescued opioid IV) iii) value of reported side effects of drug classes for this paper is unclear iv) determination of drug class hepatotoxicity is unclear 5) The paper structure and presentation may be improved (e.g. consistency with titles/subtitles, correct use of table column/row headings/titles, table name and description, optimal table page orientation etc).

Reviewer #2: 1. Please retitle and remove causal language; a cross-sectional convenience sample supports associations, not “effectiveness.”

2. State a single primary research question up front and align outcomes, analyses, and conclusions strictly to it.

3. Define mutually exclusive exposure groups (e.g., opioid-only, regional + non-opioid adjuncts, true multimodal) and list specific agents, doses, routes, and timing.

4. Normalize opioid exposure using morphine milligram equivalents (include PCA/infusions) to enable interpretable between-group comparisons.

5. Provide a sample-size/power justification, identify the primary endpoint, describe missing-data handling, and implement multiplicity control.

6. Name and cite the anxiety/depression instrument (e.g., HADS/PHQ-9/GAD-7), specify timing, and incorporate scores into adjusted analyses.

7. Clarify pain-score timing; if repeated, model trajectories (mixed effects) and discuss minimally important difference thresholds to frame clinical relevance.

8. Update and balance the literature (2023–2025), especially LMIC perioperative pain guidance and multimodal analgesia studies; discuss where your findings concur/diverge

**Do you want your identity to be public for this peer review?** For information about this choice, including consent withdrawal, please see our Privacy Policy

Reviewer #1: **Yes: ** Andile Dube

Reviewer #2: **Yes: ** Bala Nimmana

---

## [Decision Letter · Decision Letter 1]

30 Nov 2025

Multimodal Analgesia in Resource-Limited Settings: A Comparative Analysis of Postoperative Pain Management Strategies in Pakistan

PGPH-D-25-01718R1

Dear Dr Rahil,

We are pleased to inform you that your manuscript 'Multimodal Analgesia in Resource-Limited Settings: A Comparative Analysis of Postoperative Pain Management Strategies in Pakistan' has been provisionally accepted for publication in PLOS Global Public Health.

Best regards,

Barnabas Tobi Alayande

Academic Editor

Reviewer Comments (if any, and for reference):

Reviewer's Responses to Questions

**Comments to the Author**

Reviewer #1: All comments have been addressed

Reviewer #2: All comments have been addressed

publication criteria?

Reviewer #1: Yes

Reviewer #2: Yes

3. Has the statistical analysis been performed appropriately and rigorously?

Reviewer #1: Yes

Reviewer #2: Yes

4. Have the authors made all data underlying the findings in their manuscript fully available (please refer to the Data Availability Statement at the start of the manuscript PDF file)?

Reviewer #1: Yes

Reviewer #2: Yes

5. Is the manuscript presented in an intelligible fashion and written in standard English?

Reviewer #1: Yes

Reviewer #2: Yes

Reviewer #1: Comments and recommendations addressed in current version of submitted document. Accept unreservedly.

Reviewer #2: The revised Methods section is clearer, but it remains overly detailed; I kindly request shortening long subsections (e.g., opioid conversion, counting rules) to improve readability.

The expanded exposure definitions are helpful.

Please add a brief clarification in the limitations regarding the absence of validated anxiety/depression tools, as this affects interpretation.

Overall its a good manuscript . All the best.

**Do you want your identity to be public for this peer review?** For information about this choice, including consent withdrawal, please see our Privacy Policy

Reviewer #1: No

Reviewer #2: **Yes: ** Bala Nimmana
